

# Planetary boundary layer height from CALIOP compared to radiosonde over China

Wanchun Zhang[1], Jianping Guo[1], Yucong Miao[1,2], Huan Liu[1], Zhengqiang Li[3], Panmao Zhai[1]

[1]State Key Laboratory of Severe Weather, Chinese Academy of Meteorological Sciences, Beijing 100081, China
[2]Department of Atmospheric and Oceanic Sciences, Peking University, Beijing 100871, China
[3]State Environmental Protection Key Laboratory of Satellites Remote Sensing, Institute of Remote Sensing and Digital Earth of Chinese Academy of Sciences, Beijing 100101, China

*Correspondence to*: Jianping Guo (jpguocams@gmail.com) and Panmao Zhai (pmzhai@cma.gov.cn)

**Abstract.** The accurate estimation of boundary layer height is key to air quality prediction, weather forecast and so on. The planetary boundary layer height (PBLH) retrieval from CALIOP is expected to complement the ground-based site measurement due to its large spatial coverage. To such end, we estimated PBLHs from CALIOP, using the combination of Haar wavelet and maximum variance techniques, which was validated against PBLHs from ground-based lidar at Beijing and Jinhua. Comparison between ground-based and satellite lidar shows good agreement with a correlation coefficient of 0.59 in Beijing and 0.65 in Jinhua. The PBLH climatology from CALIOP was compiled over China during 2011 to 2014. Maximum PBLH was seen in summer as compared to lower value in other seasons. Prior to intercomparisons between CALIOP- and radiosonde-derived PBLHs, three matchup scenarios were proposed according to the position of each radiosonde site relative to its closest CALIPSO ground tracks. The CALIOP observations belonging to Scenario 2 were found to be better





for comparison with radiosonde-derived PBLH, owing to smaller difference between them. The PBLHs at early summer afternoon range from 1.6 km to 2.0 km, accounting for over 70% of the total radiosonde sites. Overall, CALIOP-derived PBLHs seem to be well consistent with radiosonde-derived PBLHs. To our knowledge, this study is the first intercomparison study of PBLH over large scale using the radiosonde network of China, shedding important light on the data quality of initial CALIOP-derived PBLH results.

## 1. Introduction

The planetary boundary layer (PBL), the lowest layer of troposphere closest to the surface, is directly influenced by the presence of the Earth's surface, and responds to surface forcings (e.g. sensible heat flux, mechanical drag) on a timescale of about an hour or less (Stull, 1988). The terrestrial PBL is extremely complex, given the nonlinearity and complexity of convective and turbulent processes occurred within PBL. The PBL processes play significant roles in modulating the exchange of momentum, heat, moisture, gases, and aerosols between the Earth's surface and the free troposphere (Hu et al., 2010, 2014; Miao et al., 2015). Therefore, there is general agreement that understanding and predicting weather, climate and air quality depend on accurate characterization of boundary layer processes and structure (Hu et al., 2010; Hong et al., 2006; Zhang et al., 2007; Medeiros et al., 2005).

The depth (or height) of PBL determines the vertical extent of turbulent mixing and convection activity within it, is a key length scale in weather, climate, and air quality models to parameterize the vertical diffusion, cloud formation, pollutant deposition (Hu et al., 2006; Seibert 2000; Xie et al., 2012). The PBL height (PBLH) typically varies from less than one hundred meters to several thousand meters (Hennemuth and Lammert, 2006). The most common PBLHs are derived from radiosonde soundings of



temperature, humidity, and so on. The balloons are required to be launched twice a day for the purpose of operational weather forecast, or 4-8 times daily from the perspective of scientific research during intensive observation period (Seibert, 2000; Liu and Liang, 2010). Although the radiosonde can provide height-resolved temperature and humidity profiles for accurate estimation of PBLH, which is independent of cloud cover conditions, it is still too sparse to detect the PBL evolution over large spatial scale, and thus can not adequately serve the PBL research on global or even regional scales (Sawyer and Li, 2013). With the limited available radiosonde observations (most from the Unite States and Europe), Seidel et al. (2010; 2012) constructed a general picture of PBLH climatology on a global scale, however, partly for the lack of observation in China, they did not give much detailed information of PBL over China. In 2011, a land-based radiosonde network across China was deployed by the China Meteorological Administration (CMA), which provides a unique opportunity to fill in the gap left.

In addition to the land-based radiosone observations, the lidars that allow the measurement of aerosol or trace gas profiles, also can be used to study PBL structure (ref). It is well known that aerosol concentrations vary significantly with height, which not only affects the detection of boundary layer, but also may be a large source of uncertainty particularly for satellite-based aerosol retrievals using wavelength of ultraviolet (UV) (e.g., Torres et al., 1998, 2013; Huang et al., 2015). Turning to the measurements of active remote sensing instruments, such as Cloud Aerosol LIdar with Orthogonal Polarization (CALIOP) aboard Cloud-Aerosol Lidar and Infrared Pathfinder Satellite Observations (CALIPSO) (Winker et al., 2007), aerosols can be detected and used as tracers of PBL dynamics. This is most likely due to the fact the number of aerosol particles in the PBL is often greater than that above the free troposphere (Leventidou et al., 2013). Most importantly, unlike the radiosonde measurement that only provides a "snapshot" PBL profile at a fixed site (Seibert et al., 2010), the spaceborne lidar can



obtain PBL variation over large area of interest, especially over remote regions (Jordan et al., 2010; Zhang et al., 2015).

The overpass time of CALIOP/CALIPSO is around 1330 Local Time (LT), which is almost coincident with the atmospheric sounding observations around 1400 Beijing Time (BJT) operated by CMA in the summer. In the late morning and afternoon time, when the convective boundary layer is well established, the top of convective boundary layer is often clearly characterized by the strong gradient of aerosol content, the lidar detected PBLH is generally close to the radiosonde-derived PBLH (Hennemuth) (Garratt, 1994; Seibert, 2000). Therefore, at the time of CALIOP overpasses (1330 LT), its detection seems suited to determine the convective boundary height.

As one of the first attempts to validate the CALIOP-derived PBLHs, Kim et al. (2008) carried out the intercomparison studies between PBLHs from radiosondes and CALIOP, showing high consistence between them. Among others, Ho et al. (2015) compared the marine boundary layer heights from CALIOP profiles with those from radiosonde soundings. On the other hand, large seasonal and diurnal variations in PBLHs were observed between the different methods applied to radiosonde, ground-based lidar, CALIOP observations over one site in South Africa (Korhonen et al., 2014). Although CALIOP possesses the ability to derive PBLHs over large and remote regions on a regular basis, these comparison studies were only involved in one or few sites, a comprehensive evaluation of CALIOP-derived PBLH with large scale land-based observations is still missing. In this study, the long-term CALIOP-derived PBLH over China will be validated and assessed by using the measurements of land-based radiosonde network of CMA.

The PBLH retrieval from CALIOP is expected to complement the ground-based site measurement due to its large spatial coverage. The main objective of this study are, therefore, to use nearly collocated



ground-based lidar observations to quantify the uncertainty of the CALIOP-derived PBLH and to further quantify the discrepancies between CALIOP-derived and radiasonde-derived PBLHs. The remainder of this paper proceeds as follows: the data and methods used are described in Section 2. Section 3 reports the evaluation results of CALIOP-derived PBLH using ground-based lidar measurements, and the spatial and temporal distribution of PBLH from CALIOP. Moreover, intercomparisons between PBLHs derived from CALIOP and radiosonde measurements will be performed. Last, a brief summary is given in Section 4.

## 2. Data and methods

### 2.1 Radiosonde observations and their processing

The radiosonde measures vertical profiles of temperature, pressure, relative humidity, wind speed and wind direction, with a vertical resolution of 10 m. The soundes are operationally launched twice a day at fixed times, i.e. 0800 BJT and 2000 BJT, throughout all the radiosonde sites shown in Figure 1. Fortunately, it is required by CMA to be increased to four times a day in summer (the flood season), i.e., 0200 BJT, 0800 BJT, 1400 BJT, and 2000 BJT to better serve the severe weather forecasting. Owe to our focus on the convective PBL in the daytime, the added 1400 BJT soundings therefore allow us to determine PBLHs over all sites throughout China for comparing with CALIOP-derived PBLHs, which is typically available at 1330 LT.

As summarized in Seidel et al. (2010), there are seven commonly used methods to derive PBLHs using the profiles of temperature, potential temperature, virtual potential temperature, relative humidity,



specific humidity, and refractivity. The traditional approach encountered in the textbooks (e.g., Oke, 1988; Sorbjan, 1989; Garratt, 1992) typically defines PBLH as the pressure level where the maximum vertical gradient of potential temperature occurs, indicative of a transition from a convectively less stable region below to a more stable region above. Recently, a more sophisticated method (Brooks, 2003; Davis et al., 2000) involves the wavelet covariance transform. The wavelet covariance transform was first proposed by Gamage and Hagelberg (1993) as a way to detect step changes in a signal. Combining the methods of wavelet covariance transform and simulated annealing (Steyn et al., 2009), Sawyer and Li (2013) developed a novel algorithm to derive PBLH from both radiosonde and lidar measurements, showing a good agreement. As such, this methods of Sawyer and Li (2013) was used in this study.

As shown in Figure 1, the observations of 113 radiosonde sites (black dots) during 2011-2014 are used to calculate PBLHs, and compared with the CALIOP-derived PBLHs.

## 2.2 Ground-based lidar observations

Ground-based lidar observations from two sites, i.e., Beijing and Jinhua, were also used to evaluate the PBLHs retrieved from CALIOP. The site of Beijing (40.00°N, 116.38°E) is located at the Institute of Remote Sensing and Digital Earth, Chinese Academy of Sciences, where the CE370 micro-pulse Lidar (made by CIMEL of France) was deployed during the period of January 1, 2014 to December 31, 2014. The profiles of aerosol backscatter coefficient obtained using CE370 have a vertical spatial resolution of 15 m. The laser transmitter system is reported to have a diameter of 20cm, which is used to expand laser beam through a refracting telescope.



The other ground-based lidar was deployed at Zhejiang Normal University (29.0°N, 119.5°E) in the urban area of Jinhua. Zhejiang Province. The altitude of this site is 71m above sea level. Jinhua is located in the Yangtze River Delta of East China, undergoing deteriorated air quality due to the rapid economic development in recent years (Guo et al., 2011).

The ground-based lidar deployed at Jinhua are similar to CALIOP with two orthogonally polarized channels at 532 nm and one channel at 1064 nm. The algorithms in Zhang et al. (2015) are applied on the profiles of ground-based lidars deployed at Beijing and Jinhua, respectively. To be more specific, the profiles corresponding to segments of CALIPSO ground track within a circle of 75km radius centered at the ground-based lidar site are included in the PBLH retrievals.

The lidar observations are scheduled paused during midday in summer to protect the optics from intense sunlight, leading to unwanted breaks of PBLH detections. The other unfavorable weather conditions (e.g. rains) can also cause the unwanted breaks in the lidar observations.

*2.3 CALIOP observations and their processing*

The CALIOP onboard the CALIPSO platform (flying as part of the A-Train satellite constellation since April 2006) is a three-channel elastic backscatter lidar, which is optimized for aerosol and cloud profiling. It measures attenuated backscatter coefficient at a resolution of 1/3 km in the horizontal and 30 m in the vertical at the visible wavelength (532 nm) and near-infrared wavelength (1064 nm) in low and middle troposphere, along with polarized backscatter in the visible channel(Winker et al., 2009).

All satellites of the A-train constellation are in a 705-km sun-synchronous polar orbit between 82 °N and 82 °S with a 16-day repetition cycle, with a nominal ascending node equatorial crossing time of



1330 (0130) local day (night) time (Liu et al., 2009; Winker et al., 2007; Winker et al., 2003). As shown in Figure 1, red lines represent the ground tracks over China for the daytime overpasses of CALIPSO (in ascending mode), while blue lines ground tracks for nighttime overpasses of CALIPSO (in descending mode). The neighboring ground track is at a longitudinal interval of approximately 150-km, varying with latitudes.

The PBLH is predominantly estimated from the CALIOP Level 1 product: the total attenuated backscatter coefficient, in combination with and Level 2 product of cloud layer products (1/3 km in the horizontal) for cloud screening. This is because that all the PBLH retrievals are limited to cloud-free scenes. According to the summary of methods used to derive PBLHs by Jordan et al. (2010), we relied on the maximum variance algorithm to derive PBLHs from CALIOP attenuated backscatter coefficient profiles at wavelength of 532 nm, in combination with the Haar wavelet technique.. The maximum variance algorithm is originated from the ideas proposed by Melfi et al. (1985) and heavily relies on the existence of a strong aerosols concentration gradient at the top of the PBL, which can be detected by examining the levels where the maximum standard deviation occurs of lidar backscatter. This method has been widely used to derive PBLHs from CALIOP so that the global seasonal variations can be inferred (McGrath-Spangler and Denning, 2012, 2013). To make the comparison of PBLHs between radiosonde and CALIOP more reliable and robust, the combined algorithm has been applied on the profiles corresponding to segments of CALIPSO ground track within a circle of 75km radius centered at each radiosonde site. All the comparisons are limited to daytime measurements due to the nature of convective boundary layer, unless noted otherwise.

Due to the most likely blocking and attenuation caused by optically thin or thick clouds, we have to perform the cloud-screen procedures prior to the algorithm mentioned above operating on the CALIPSO





level 1 profile data. The data were retained for grid points where the number of valid CALIPSO overpasses exceeded 15% of the total number of overpasses. As such, we can minimize the effect of clouds on the retrieved PBLHs to a certain degree. Note that over regions where BL is not convective the retrieved values are not representative of the PBLH (Liu and Liang, 2010).

5    As a good case in point for a better view of the results derived using the above algorithms, the CALIOP-derived PBLHs (indicated by the black line) on 15 January 2011 over southeastern China is shown in Figure 2. To improve the signal-to-noise ratio (SNR) to derive the boundary layer top, 17 profiles at 333-m resolution along track were resampled to one 5-km resolution profile. By visual interpretation, we can see that derived-PBLHs locate accurately on the boundary where aerosol 10    backscatter signals changes abruptly, indicating that the combined algorithms is reliable.

## 3.  Results and discussion

### 3.1 Evaluation of CALIOP-derived PBLH against ground-based lidar-derived PBLH

As a first attempt to perform comparison between PBLHs from different sources, CALIOP-derived results have to undergo an evaluation using ground-based lidar, which typically shares the similar 15    techniques. To minimize the influence of cloud on the PBLH determination from lidar, we exclude all the lidar measurements of Beijing and Jinhua with cloud cover.

Figure 3 shows that the scatter plots of the ground-based lidar derived PBLHs versus CALIOP-derived PBLHS over Jinhua (29.1°N, 119.6°E) and Beijing (40.0°N, 116.4°E). Due to the twice-per-month revisit period of CALIPSO satellite, only 17 cases out of 24 at Beijing are selected, in which 20    both CALIOP and ground-based lidar have simultaneous measurements at 1330 LT. And the PBLH retrieval has been carried out for 7 cases out of 12 at Jinhua. For the overall comparison between the





PBLHs derived from ground-based lidar and CALIOP, the correlation coefficient through orthogonal regression reaches 0.59 at Beijing and 0.65 at Jinhua, respectively, which shows a good agreement. Similar correlation coefficient between the ground-based lidar and CALIOP derived PBLHs has been reported at SACOL site of northwestern China (Liu et al., 2015).

*3.2 CALIOP-derived PBLH Climatology throughout China*

Figure 4 presents the spatial distributions of seasonal mean PBLHs with $0.2^o \times 0.2^o$ resolution derived from CALIPSO afternoon measurements during the period 2011 through 2014. The original 5 km PBLH data have been smoothed and resampled to 20 km resolution to highlight the coherent large-scale structures. It can be clearly seen that the PBLHs over China exhibit large spatial and seasonal variations.

On average, both Figure 4 and Table 1 indicate that the highest PBLs ($1.82km \pm 0.31km$) were developed in summer (June, July and August), mainly ranging from 1.5 to 2.5 km. On the other hand, the lowest PBLH values and variability ($1.51km \pm 0.40km$) were occurred in winter (December, January and February). This is most likely due to that the development of PBL is directly caused by the surface thermal and mechanical forcings. In summer, the intense solar radiation favors the development

of PBL (Stull et al., 1988).

In terms of the discrepancy in spatial distribution of PBLH, the Tibetan Plateau (TP) was characterized by high values, irrespective of the evolution of seasons. Over eastern China, particularly the regions with large population and severe air pollution (Guo et al., 2009; 2011) (e.g. North China Plain, the Yangtze River Delta, and Pearl River Delta), the PBLHs was higher in spring and summer,

but did not show expected large seasonal variation. During the season haze frequently occurs, the aerosol particles within the development of PBL may be suppressed by aerosol radiative effects (Xia et al., 2007), and leads to a relatively shallow PBLH (Quan et al., 2013; Miao et al, 2016; Gao et al. 2015).



The spatial distribution of PBLH revealed a tendency for higher PBLH over high elevation regions, similar distribution in the United States had been reported by Seidel et al. (2012). Such spatial variation of PBLH may be related to the local land surface and hydrological processes (Seidel et al., 2012).

### 3.3 Matchup between CALIOP profiles and radiosonde soundings

Due to the neighboring ground tracks of CALIPSO at approximately 100-150 km longitudinal interval over China, a 75km radius circle centered at each radiosonde site was determined for the matchup of CALIOP and radiosonde site. As revealed in Section 2.3, the PBLHs derived from the profiles were averaged for comparison, which correspond to segments of CALIPSO ground track within a 75-km-radius circle centered at each radiosonde site. After multiple rounds of iteration through the positions of

each radiosonde site over China relative to its closest CALIPSO ground tracks, a total of three scenarios can represent all the cases, as shown in Figure 5. Scenario 1 denotes the cases with two CALIOP ground tracks, the shortest distance to which each is more than 37.5km from each radiosonde site. In contrast, Scenario 2 represents the cases with one CALIOP ground track, the shortest distance to which is less than 37.5km from each radiosonde site. On the other hand, Scenario 3 is the same as Scenario 2

except for the shortest distance to which is more than 37.5km from radiosonde site.

   The details of classification criteria can be summarized in Table 2. Out of the total of 113 radiosonde sites were classified, 64 sites belonged to Scenario 2. That means about 56.6% of all radiosonde sites make a good match with CALIOP profiles for its nearest distance to CALIPSO ground tracks less than 37.5km. By comparison, there are 22 sites (19.5%) attributed to Scenario 1 whereas 27 sites (23.9%)

scenario 3.





Figure 6 shows the geophysical distribution concerning the location of radiosonde sites relative to its closest CALIOP ground tracks inside a circle of radius 75 km over China, which are stratified by Scenarios 1, 2, and 3. Owing to the nearest distance to radiosonde site in Scenario 2, profiles in CALIOP observations can be used to better capture the PBL evolution, and thus facilitate the intercomparisons. It happens that the radiosonde sites (56.6%) belonging to Scenarios 2 are uniformly distributed over China, indicating that most of the radiosonde sites in China can be collocated well with afternoon CALIPSO overpass.

Interestingly, the radiosonde sites for Scenario 1 are basically located in the northern China, as opposed to those for Scenarios 3 in the southern China. The distinct discrepancy in geographic distributions of radiosonde sites belonging to Scenarios 1 and 3 are most likely due to the latitude differences. The more northward the radiosonde sites, the more frequently the CALIPSO overpasses over the same circle of 75 km radius. More importantly, because the region of interest (China) spans several time zones, the spatial variations of radiosonde-derived PBLHs observed at fixed observation times (1400 BJT) tend to be conflated with diurnal variations, as discussed in the following Section 4.

## 3.4 Intercomparison between CALIOP- and radiosonde-derived PBLHs

Using the algorithms as detailed in Section 2, the PBLHs at all the 113 radiosonde sites have been successfully derived, so have the CALIOP-derived PBLHs. According to three matchup scenarios for both CALIOP profiles and radiosonde sites described above, the difference of PBLH from CALIOP 1330 LT minus that from radiosonde observations at 1400 BJT in the summertime (June-July-August) during the period of 2011-2014.



As shown in Figure 7(a), most of the radiosonde sites to the east of 110 $^{o}$E longitude exhibit negative values, indicating CALIOP-derived PBLHs tend to be underestimated compared with radiosonde-derived PBLHs. In contrast, it is a different story (to be overestimated as compared with radiosonde) for the sites to the west of 110 $^{o}$E longitude, especially in provinces such as Xinjiang, Sichuan and

Chongqing. Because observation time for CALIOP corresponds to 1330 LT in the western China while late afternoon in the east, the radiosonde at 1400 BJT in the west are expected to be in association with weak afternoon convection, leading to relative low PBLHs derived from radiosondes. Note that we cannot totally rule out other factors that may also contribute to the east-west gradient. However, there are other aspects neglected to be discussed here, which are likely to be contributed to the discrepancies

between the two methods.

We divided all sites in Figure 7 (a) into three subgroups according to the matchup scenario described in previous section, as shown in Figure 7 (b-d). From the perspective of PBLHs over any radiosonde site, CALIOP-derived PBLHs tend to be underestimated compared with radiosonde-derived PBLHs, born out by the results in Table 2 and Figure 8 due to the larger percentages of sites (77 of 113 sites, i.e.,

68%) showing lower PBLH values. In terms of mean biases between CALIOP- and radiosonde-derived PBLHs, Scenario 2, as expected, has smaller magnitude (0.17 km), as compared with Scenario 1 (with a magnitude of 0.22 km). On the other hand, the smallest mean bias (0.15 km) was observed for Scenario 3.

Figure 8 shows the occurring frequency for the radiosonde sites as stratified by binned radiosonde-

derived mean PBLHs (1400 LT) and CALIOP-derived mean PBLHs (around 1330 LT) over China in the summertime (June-July-August) during the period of 2011-2014. Typically speaking, the PBLHs at early summer afternoon over China range from 1.6 km to 2.0 km, accounting for over 70% of the total



radiosonde sites. The pattern in Figure 8(c) is more similar to that in Figure 8 (a), suggesting that the results from Scenario 2 to some extent are representative of the overall results over all sites. As such, comparison of the histogram of CALIOP PBLHs to the radiosonde observations indicates that they are in good enough agreement with each other.

## 4. Conclusions

This study presents initial validation results of space-borne CALIOP-derived PBLHs by comparing with coincidental observations from two ground-based lidars at Beijing (January 1, 2014 to December 31, 2014) and Jinhua (June 1, 2013 to December 31, 2013). Results show that the correlation coefficient is about 0.59 in Beijing and 0.65 in Jinhua, respectively. The selected data set represents two different underlying land surfaces, i.e., urban and mountain area, both of which are obtained under cloud-free conditions.

The climatology of seasonal mean PBLHs at $0.2^{o} \times 0.2^{o}$ resolution has been constructed, as derived from daytime CALIPSO measurements during the period 2011 through 2014. The PBLHs over China are found to exhibit large spatial and seasonal variations. On average, summer (June, July and August) is characterized by the highest PBLH values, as opposed to the lowest PBLH values in winter (December, January and February). Such seasonal variation of PBLH may be caused by the seasonal variation of solar radiation.

Prior to comparing CALIOP-derive PBLHs with radiosonde, three matchup scenarios are proposed according to the position of each radiosonde site over China relative to its closest CALIPSO ground tracks, which cover all the collocated data-pairs of CALIOP and radiosonde. Matchup maps for



Scenario 2 indicate that most of the radiosonde sites in China can be collocated very well with afternoon CALIPSO overpass. As such, the profiles in CALIOP observations belonging to Scenario 2 seem better for comparison with radiosonde-derived PBLH, owing to smaller difference between them.

Overall, CALIOP-derived PBLHs tend to be underestimated compared with radiosonde-derived PBLHs. On the other hand, the PBLHs at early summer afternoon over China mostly range from 1.6 km to 2.0 km, accounting for over 70% of the total radiosonde sites. Therefore, CALIOP PBLHs seem to agree pretty well with radiosonde-derived PBLHs. Despite the limitation in the presence of clouds, CALIOP has been routinely available for determination of PBLHs and therefore are a valuable method for long-term climatology analyses. To our knowledge, this study is the first intercomparison study of PBLHs between CALIOP- and radiosonde-derived PBLHs over large scale using the radiosonde network of China, although much detailed regional analyses have not been dealt with, which merit further investigation in the near future.

## Acknowledgements

This study was financially supported by the National Natural Science Foundation of China (Grant 91544217), Ministry of Science and Technology of China (Grant no. 2014BAC16B01), Natural Science Foundation of China (Grant 41471301) and Chinese Academy of Meteorological Sciences (Grant 2014R18). The authors would like to acknowledge CMA for providing the radiosonde dataset for us to use. Special thanks go to NASA for making the CALIOP products accessible for public use, Anhui Institute of Optics and Fine Mechanics (AIOFM) and Institute of Remote Sensing and Digital Earth of



Chinese Academy of Sciences, Chinese Academy of Sciences (CAS) for providing the ground-based lidar data.

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



## Table list:

5 **Table 1.** Descriptions regarding the statistical results of seasonal mean PBLH estimated from CALIOP.

|  | Spring | Summer | Autumn | Winter |
|---|---|---|---|---|
| **Maximum PBLH (km)** | 4.57 | 4.40 | 3.60 | 6.13 |
| **Minimum PBLH (km)** | 0.15 | 0.38 | 0.22 | 0.21 |
| **Mean PBLH (km)** | 1.72 | 1.82 | 1.56 | 1.51 |
| **Standard deviation PBLH (km)** | 0.35 | 0.31 | 0.30 | 0.40 |

**Table 2.** Detailed descriptions with regard to the classification criteria of scenario of the positions of radiosonde site relative to the closest CALIOP profiles, including the number of CALIPSO ground tracks for each scenario, the shortest distance (D) to ground tracks, the total number of sites for each scenario in China, as well as the number of sites with overestimated averaged PBLHs (O) or underestimated averaged PBLHs (U) from CALIOP compared with radiosonde.

| Scenario | # of CALIPSO ground tracks | D (km) | # of sites | # of sites with O | # of sites with U |
|---|---|---|---|---|---|
| 1 | 2 | $37.5 < D \leqslant 75$ | 22 | 11 | 11 |
| 2 | 1 | $0 \leqslant D \leqslant 37.5$ | 64 | 18 | 46 |
| 3 | 1 | $37.5 < D \leqslant 75$ | 27 | 7 | 20 |

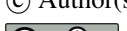



# Figure list

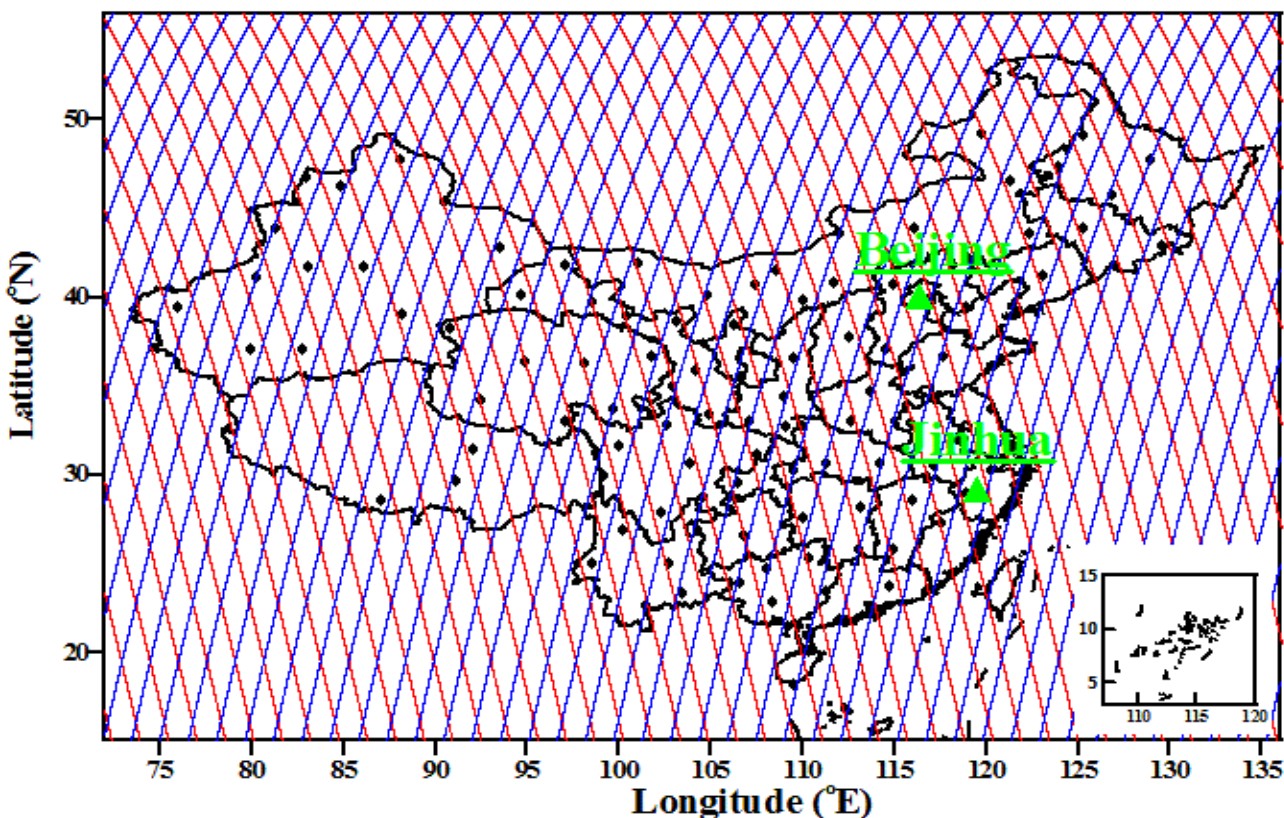

**Figure 1.** Geographic distribution of radiosonde sites and ground tracks for CALIPSO over China. Red
lines represent the ground tracks for the CALIOP daytime orbits (in ascending mode), while blue lines
for the CAILOP nighttime orbits (in descending mode). The black dots denote all radiosonde sites
operated and maintained by China Meteorological Administration. Beijing and Jinhua (green solid
triangles) are two sites deployed with ground-based lidar.

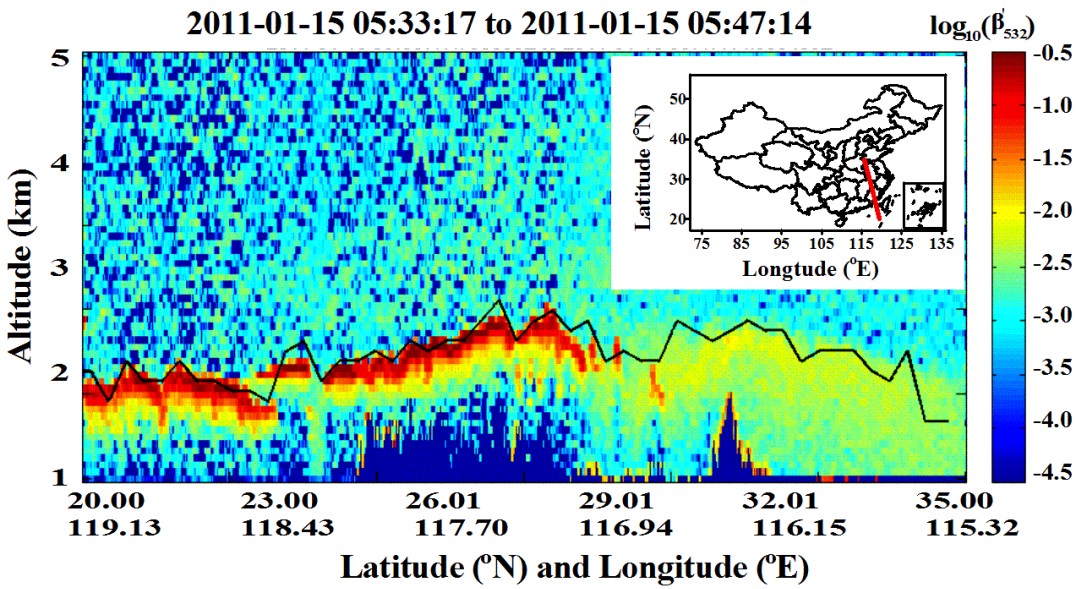

**Figure 2.** Curtain plot of attenuated backscatter coefficient as observed from CALIOP aboard CALIPSO on 15 January 2011. The black line indicates the derived PBLH and the blue region represents the terrain surface. The red line in the inlet map corresponds to the ground track of CALIOP/CALIPSO over southeastern China.





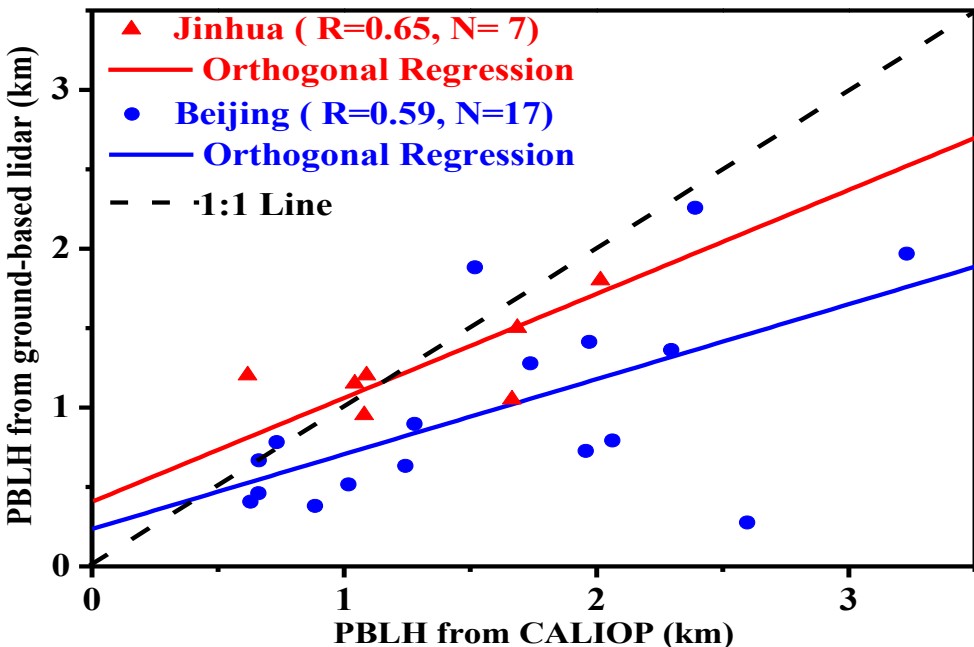

**Figure 3.** Scatter plot for comparing PBLHs from CALIOP to those from ground-based lidars at Beijing (blue dots) during the period January 1, 2014 to December 31, 2014 and Jinhua (red triangles) during the period of June 1, 2013 to December 31, 2013. Blue and red lines denote the linear fit to the data at Beijing and Jinhua sites, respectivley, and black dash line the 1:1 correlation. The number of collocated data samples and corresponding correlation coefficient(R) are shown as well.





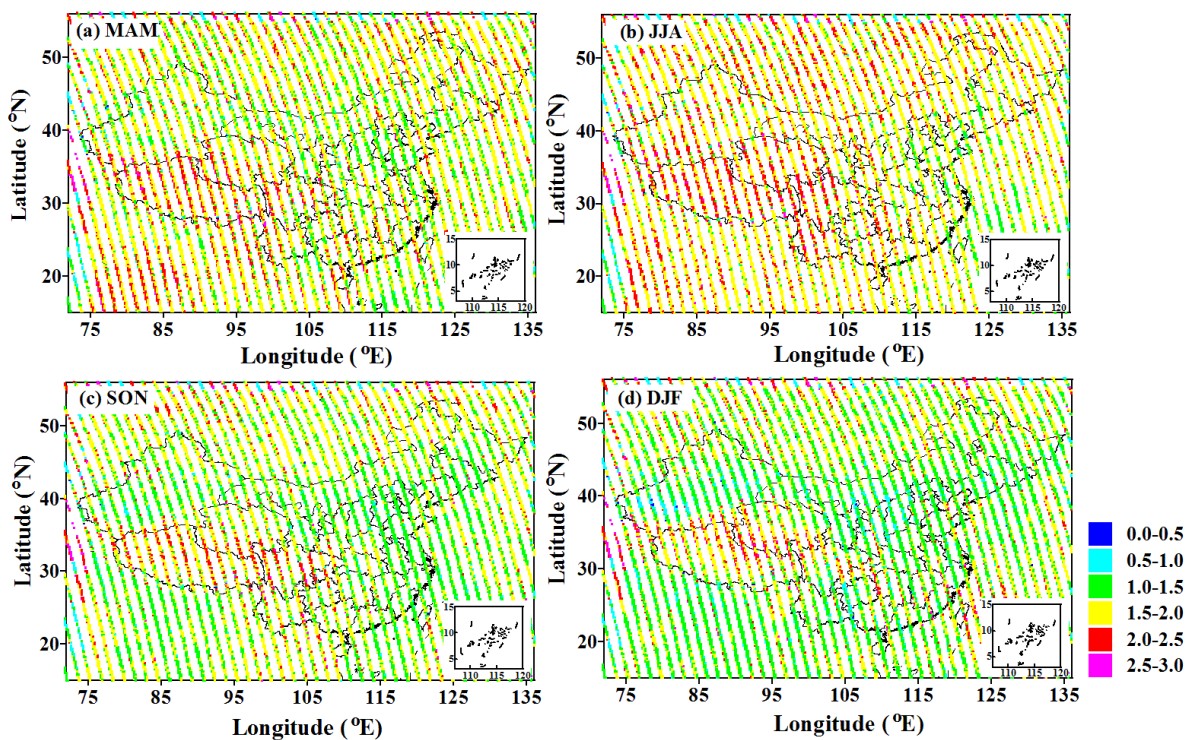

**Figure 4.** Spatial distributions of mean PBLH climatology derived from CALIOP at 1330 BJT in (a) spring (March-April-May, MAM), (b) summer (June-July-August, JJA), (c) autumn (September-October-November, SON) and (d) winter (December-January-February, DJF) during the period 2011 - 2014. Horizontal resolution is resampled to 20 km along the ground track.





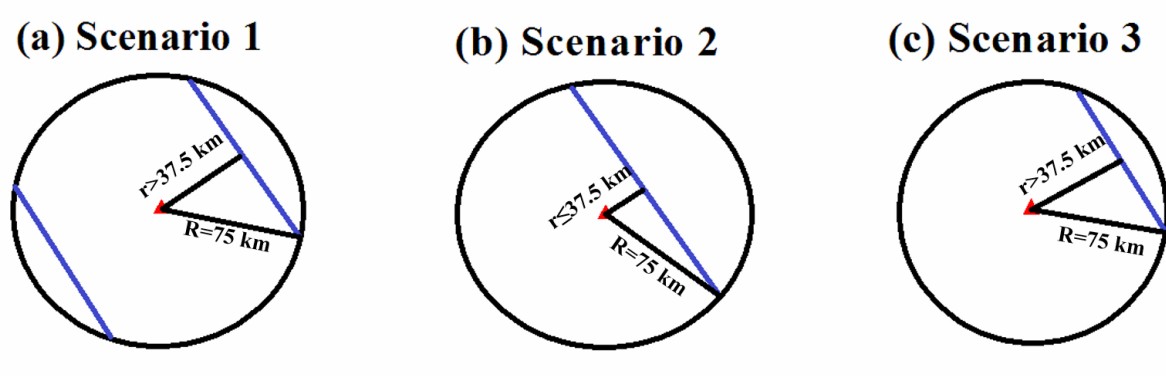

**Figure 5.** Schematic diagrams s showing the location of CALIOP ground tracks relative to radiosonde sites according to (a) Scenario 1 (with two CALIOP ground tracks, the shortest distance to which each is more than 37.5km from radiosonde site); (b) Scenario 2 (with one CALIOP ground track, the shortest distance to which is less than 37.5km from radiosonde site; (c) Scenario 3 (with one CALIOP ground track, the shortest distance to which is more than 37.5km from radiosonde site) showing the geometric relationship of CALIOP ground tracks relative to radiosonde sites. A circle with a radius of 75 km centered at radiosonde sites was chosen to obtain averaged PBLH from CALIOP, as compared with the measured PBLH from ground-based soundings.





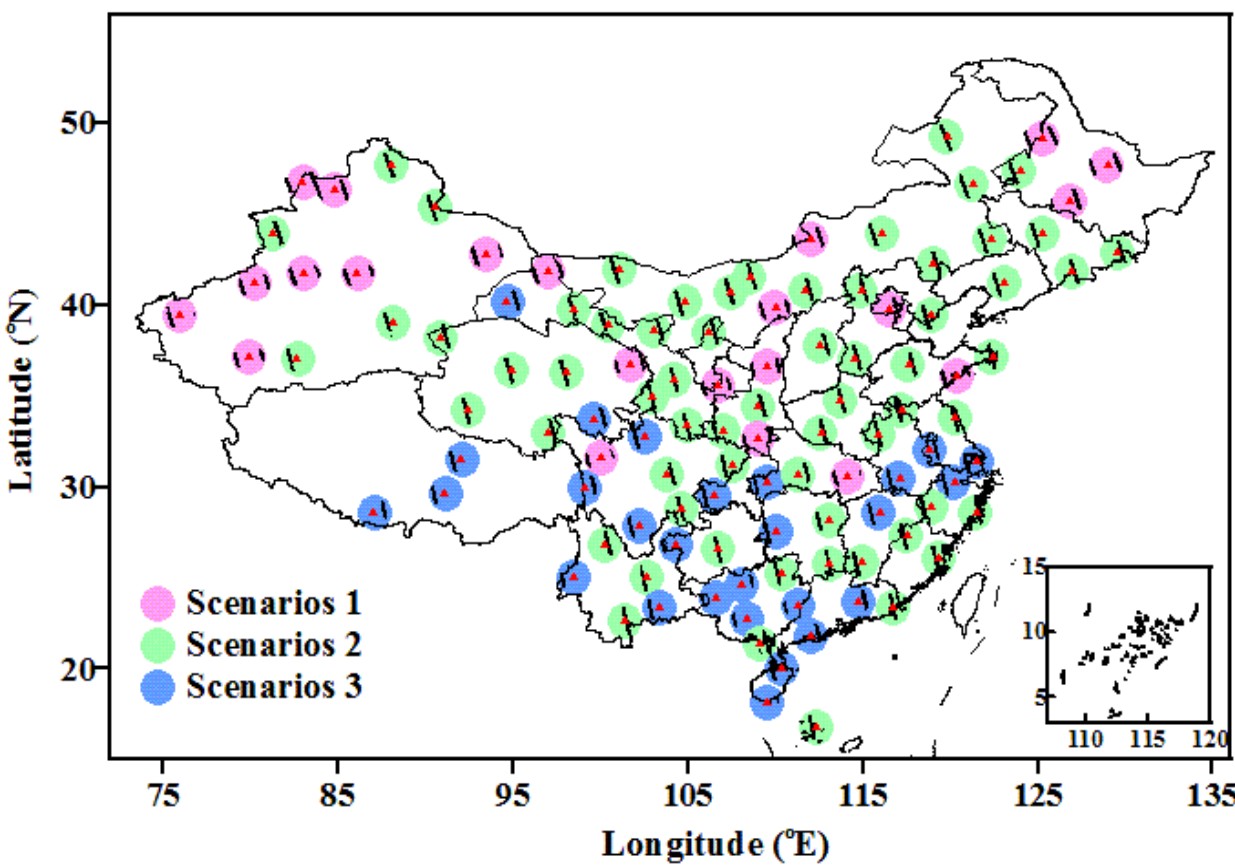

**Figure 6.** The geographic map showing the location of radiosonde sites relative to CALIOP ground tracks over China. The red triangles denote the radiosonde sites, and the black lines show CALIOP tracks chosen for comparisons. The solid circles in cayon, green and blue correspond to Scenarios 1, 2, and 3 defined in Figure 5.

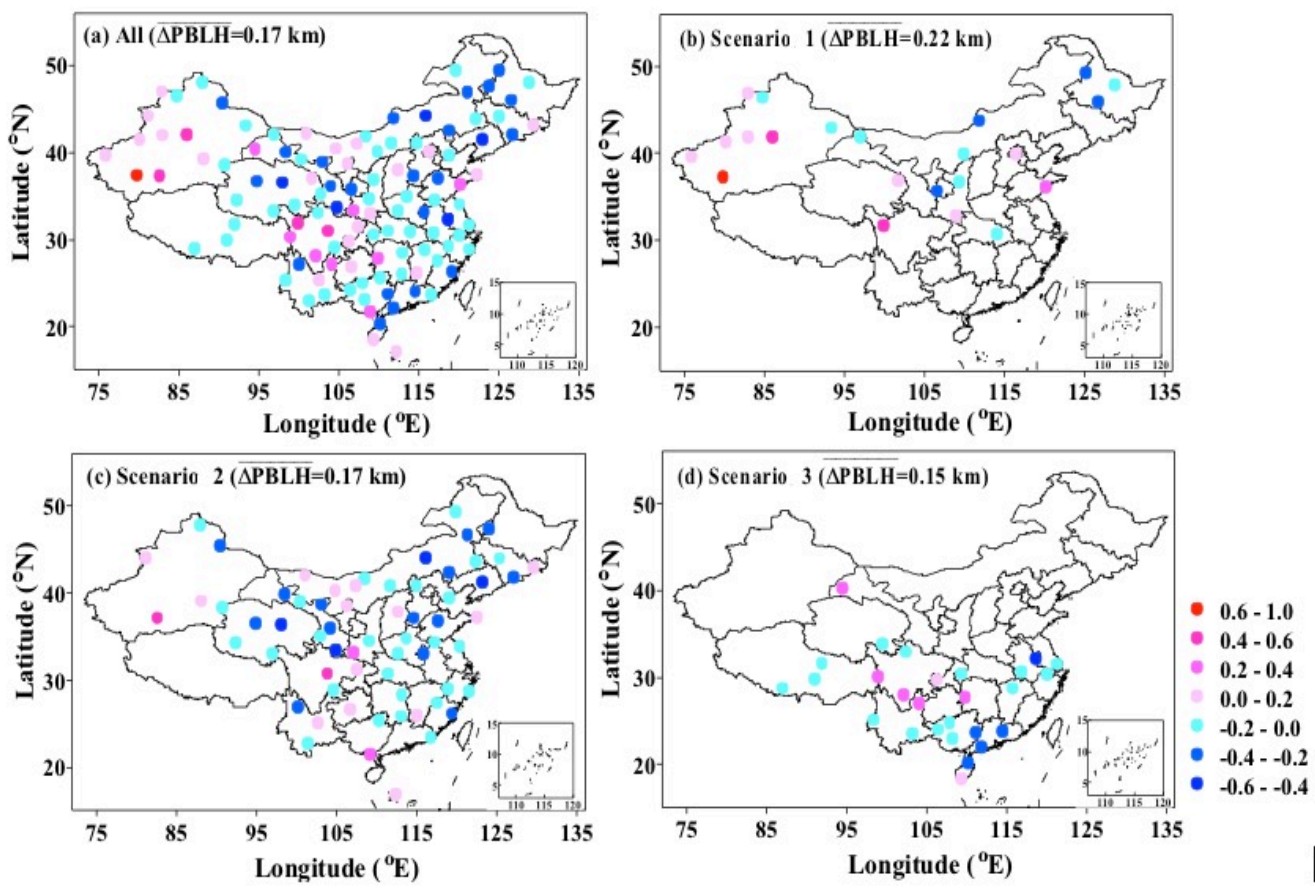

**Figure 7.** The geophysical distribution map concerning the difference of PBLH derived from CALIOP 1330 LT minus that derived from radiosonde observations at 1400 BJT in the summertime (June-July-August) during the period of 2011-2014. The differences of PBLHs are shown for all radiosonde sites in China (a), the radiosonde sites belonging to Scenario 1 (b), Scenario 2 (c), and Scenario 3 (d), respectively.





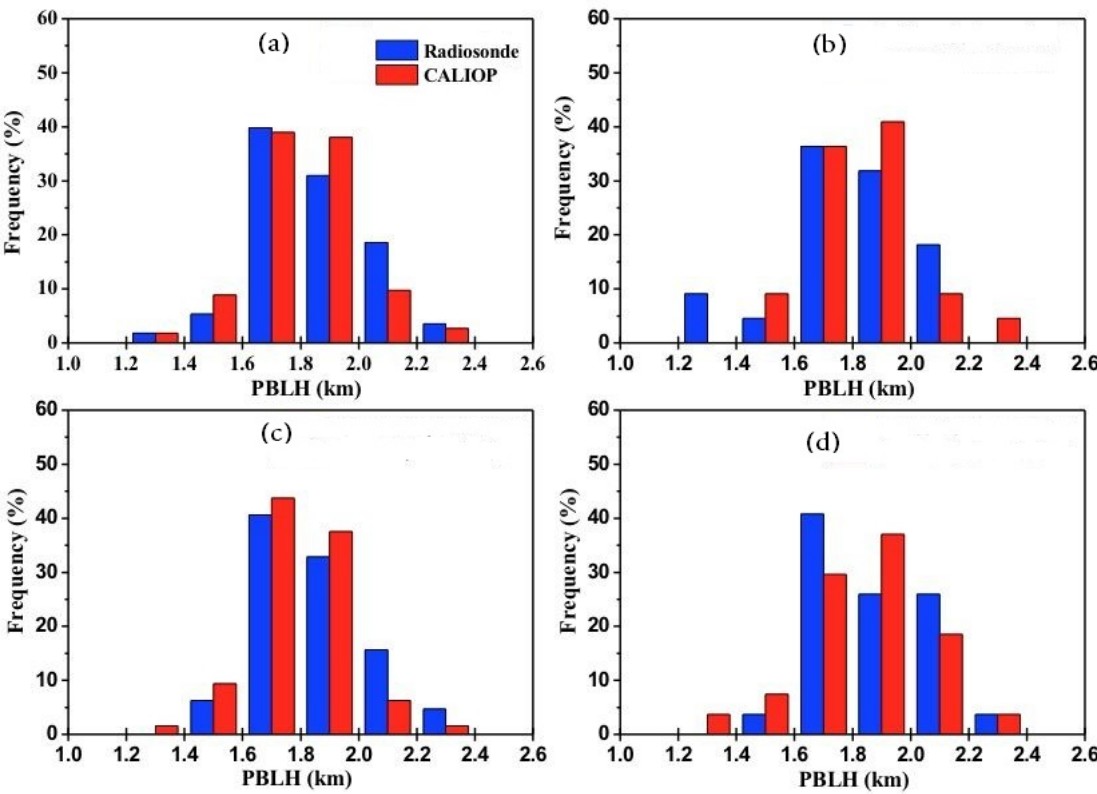

**Figure 8.** Histogram of the number of radiosonde sites, stratified by binned radiosonde-derived mean PBLHs (blue bars, 1400 LT) and CALIOP-derived mean PBLHs (red bar, around 1330 LT) over China in the summertime (June-July-August) during the period of 2011-2014 for all radiosonde sites (a), the radiosonde sites belonging to Scenario 1 (b), Scenario 2 (c), and Scenario 3 (d), respectively. The frequency is calculated as the ratio of the number of radiosonde site in each PBLH bin to the total number of radiosonde sites.