# Peer review of "Planetary boundary layer height from CALIOP compared to radiosonde over China"

_Atmospheric Chemistry and Physics, 2016_

## Referee Comment (RC1) · Anonymous Referee #2 · 2 May 2016

**Comments on "Planetary boundary layer height from CALIOP compared to radiosonde over China"**

**General Comments**

The planetary boundary layer height (PBLH) is an important length scale in weather, climate and air pollution models. The CALIOP-derived PBLHs can construct the PBLH climatology on a global scale. The problem is that the validity of CALIOP-derived PBLH should be examined and the uncertainties of CALIOP-derived PBLH should be known. In this paper, the authors compared the CALIOP-derived PBLH to the radiosonde-derived PBLH in China. The results suggest that they agree very well. The authors also analyzed the difference in the PBLHs derived from the two methods, and showed the spatial distribution of deviations. The results in this paper can help to understand the applicability of CALIOP-derived PBLH in China, and provide the basic information for further investigations. However, some details of the dataset should be further specified, and the English writing should be further improved. Therefore, I recommend the manuscript for publication in ACP, pending minor revisions.

**Specific Comments**

1. The author declare that the method of Sawyer and Li (2013) was used in this study (in page 6 line 9-10). I suggest that the authors should give a concise introduction of this method, so that the readers can understand how the PBLH is derived from CALIOP in this paper rather than the cited paper. Is this method also applied to the radiosonde data to derive the PBLH? Because the measurement time is almost at noon, the potential temperature profile should exhibit the typical structure of convective BL. Thus the method of maximum potential temperature gradient is suitable for determining the PBLH. Why not use the maximum gradient method? The authors should explain the reason.

2. The derived PBLH should be the height above the ground. However, shown in Fig. 2, the derived PBLH is above the sea level. Is the terrain height derived from CALIPSO or obtained from other data source? The authors should specify this issue. As shown in Fig. 2, the terrain surface is not very clear in some places.

3. In page 9 lines 3-4, the authors state "Note that over regions where BL is not convective the retrieved values are not representative of the PBLH (Liu and Liang, 2010)". Also in this section (Section 2.3), the authors describe the method how to eliminate the effects of clouds on the CALIOP-derived PBLH. In other words, the CALIOP data in clear days are used to derive the PBLH, and the BL should be convective. Moreover, the passing time of CALIPSO is 13:30 BJT. Thus it can be expected that the PBLH at this time is not very low. However, Table 1 shows that the minimum PBLHs in different seasons are 0.2-0.4 km. I think these values are unbelievable. On the other hand, Table 1 show that the maximum PBLHs in different seasons are 4-6 km with the largest value in winter. I think these values

are also unbelievable. It is likely that uncertainties are introduced in the CALIOP-derived PBLH. Then the problem, to what extent the CALIOP-derived PBLH over China is reasonable, arises. I suggest the author discuss this problem and provide additional information about the statistics of the CALIOP-derived PBLH. For example, by setting the reasonable range of PBLH based on the up-to-date knowledge, the percentage of the derived PBLHs that are in this range can be calculated and compared.

4. For the title of Table 1, "seasonal mean" is not accurate. I think, the maximum PBLH, as well as the minimum PBLH, is not the seasonal mean. Maybe "Statistics of the CALIOP-derived PBLH in different seasons" is more accurate. "Standard deviation PBLH" should be "Standard deviation of PBLH". Moreover, the authors should tell the readers how to determine/calculate the values in the table. Is the maximum/minimum PBLH determined as the maximum/minimum value of one grid in the duration or as the average of the maximum/minimum values at every grid in China? Is the standard deviation calculated at every grid and then averaged in China or calculated directly using all the data?

5. Following above question, Fig. 8 shows that the CALIOP-derived PBLH ranges from 1.2 km to 2.4 km. But the statistics in Table 1 show that the CALIOP-derived PBLH varies in a very large range. How many data are not considered in Fig. 8? The authors should specify this issue in the text or in the figure caption.

6. The authors declare in the abstract "The CALIOP observations belonging to Scenario 2 were found to be better for comparison with radiosonde-derived PBLH, owing to smaller difference between them". Similar statements are found in the conclusion section. However, Fig. 7 shows that the mean difference for Scenario 3 is the smallest. What is the solid evidence for this conclusion?

**Technical Corrections**
1) The grammatical errors should be corrected (Just some are listed here. The author should thoroughly check for simple typos and grammatical errors). For example,
Page 2 line 1, "for comparison with" should be "in comparison with".
Page 2 line 2, "at early summer afternoon" should be "in early summer afternoon".
Page 3 line 20, "the fact the number" should be "the fact that the number".
Page 4 line 22, "are" should be "is".
Page 6 line 9, "this methods" should be "this method".
Page 8 line 7, "in combination with and" should be "in combination with".

2) Fig. 2, at the top of this figure the times "05:33:17" and "05:47:14" should be the local times "13:33:17" and "13:47:14".

3) Fig. 7, the value of mean difference between the CALIOP- and radiosonde-derived PBLHs in each panel (0.17km, 0.22km, 0.17km and 0.15km respectively). But the figure shows that the difference for a single site is either positive or negative (denoted by different colours). How to calculate the mean value, directly or by the absolute values? I guess by absolute values. Therefore the absolute value sign should be added to ΔPBLH.

---

## Referee Comment (RC2) · Anonymous Referee #1 · 3 May 2016

Comments on "Planetary boundary layer height from CALIOP compared to radiosonde over China"

The planetary boundary layer height (PBLH) is an important parameter for the weather and climate study, as well as atmospheric pollution study. This study tries to obtain global PBLH based on CALIPSO satellite observations, and carried out an intercomparison study with those from radiosondes and lidars here. The results suggest that they agree reasonably well in China regions. This is a valuable contribution to the science community to better understand the potential applicability of CALIPSO observations to obtain PBLH. However, this paper does need some improvement as detailed below, particularly regarding to the English writing. I would recommend the manuscript for publication in ACP, pending minor revisions.

Main Comments
(1) The English writing strongly need improve. The paper descriptions could be more concise and accurate.
(2) One key role of this study as the author expressed is "The PBLH retrieval from CALIOP is expected to complement the ground-based site measurement due to its large spatial coverage". However, I think the pass of CALIPSO satellite over a specific location is limited. May you please provide more information about the CALIPSO passed regions?
(3) Section 2.1, I would like to know the uncertainties in the PBLHs obtained from radiosondes, which is very important since the authors are using them to evaluate those from CALIOP.
(4) Section 2,2, what is the uncertainties of PBLHs from lidars, and what are the extra uncertainties caused by the selection of compare region size?
(5) Section 3.1, this is a comparison. If you would like to say 'evaluation", you need assume the accuracy of ground-based lidar-derived PBLH with at least clear uncertainty information.
(6) Section 3.2, I would suggest you add the climatology of PBLH from the readiosonde profiles over China and compare this with your results from CALIPSO observations. This could let us know how reliable of your CALIPSO-derived PBLHs.

Page1
(1) Line 12: The description could be more concise: the accurate estimation of planetary boundary layer height (PBLH) …. The PBLH retrieved from …"
(2) Line 17: ground-based and satellite-based or ground-based and spaceborne
(3) Line 17-18, for r=0.59 or 0.65, could we say "good agreement"?
(4) Line 19, 'during 2011 to 2014' -> 'for the period from 2011 to 2014'
(5) Line 19, lower values
(6) What is the uncertainty for PBLH from radiosonde observations? What are the factors that could result in the differences in PBLH between satellite- and ground-based observations, and their contributions?

Page2
(7) Line 17, how do you arrange the order of references?
(8) Line 18-20, the sentence have grammar error with 2 verbs.

(9) line 1-3, why is it required 4-8 times for IOP experiment?
(10) line 4, how accurate of the PBL height is it for the measurements from radiosondes?

(12) line 12-13, what do you mean with (ref) in these lines? Reference?

(13) line 13-15, what do you mean for this sentence: "large seasonal and diurnal variations in PBLHs were observed between the different methods applied to radiosonde, ground-based lidar, CALIOP observations over one site in South Africa"

(14) what do you mean for "large scale land-based observations"?
(15) how reliable for the ground-based lidar observation of PBLH?

(16) line 14, times -> time
(17) line 15, why call the summer as flood season? It might be wet season, but not good as flood season?
(18) line 16, what do you mean for "severe weather forecasting"?
(19) line 16-19, 'owe to …, … therefore…"?

(20) line 9, What are you comparing to regarding "a good agreement"?
(21) line 9, 'this methods of … was …'?

(22) line 6, 'the algorithm in Zhang et al. (2015) are applied on …" -> "the algorithm developed by Zhang et al. (2015) are applied to …"
(23) line 7, what kind of profiles are you talking about? lidar profiles?
(24) line 8-9, why do you choose the area with radius of 75 km?
(25) line 10-13, what are the data volume fraction for these cases?
(26) line 17-19, please correct the sentences, such as "It measures attenuated backscatter coefficients at resolutions of 1/3 km in the horizontal and
30 m in the vertical at the visible wavelength …"

(27) line 7, " in combination with and …"?
(28) line 8-9, "This is because that …", You do not need to explain since you have said for "cloud screening"
(29) line 9-11, please indicate the advantage of your choosing method.
(30) line 11, there are two periods.

(30) line 9-16, please tell readers the uncertainties or the uncertainty-influential factors for this determination method.

(31) line 16-19, this is redundant since you have mentioned the 75 km earlier. Also, why do you select 75 km, not 50 or 25 km?

(32) line 1, what do you mean "valid" here? For the overpasses, are there invalid ones? I donot understand.

(33) line 4, How do you determine if the BL is convective or not?

(34) line 5-10, you just gave one case to show the good agreement between two algorithms (even 17 profiles averaged within a 5 km region). This is not enough to conclude that "the combined algorithms are reliable".

(35) line 10, 'is' ->'are'

(36) line 13, are you sure your comparison study is "a first attempt"?

(37) line 15-16, how do you exclude the cases with cloud cover? In other words, how do you get the cloud coverage?

(38) line 17, "shows that"? I believe it should be just "shows"

(39) line 17-21, for so limited data samples, how reliable are the comparison results?

(40) line 1-2, the correlation coefficients are low, why do you say 'show a good agreement'?

(41) line 11-13, the variability in winter (0.4 km) is larger than that in summer (0.31 km), why do you say the lowest PBLH variability occurs in winter?

(42) line 13, "were occurred" -> "occur'

(43) line 14-15, please modify the description to make it more concise.

(44) line 19, 'was' -> 'were'

(45) line 21, 'may be suppressed by aerosol radiative effects and aerosol-wind interactions (Xia et al., 2007; Yang et al., 2016)'
Yang, X., C. Zhao, J. Guo, Y. Wang, 2016, JGR: intensification of air pollution associated with its feedback with surface solar radiation and winds in Beijing.

(46) line 2, 'had been' -> "have been"

(47) line 5-7, this information has been described two times earlier. I would suggest a more detailed description for only one time.

(48) line 7-9, this also seems redundant.

(49) line 14, delete "On the other hand,"

(50) line 16, 'can be' -> 'are'

(51) line 8, what do you mean for "basically"?

(52) line 11-12, could you give me a little more explanation? I do not understand the logic here.

(53) line 16-17, "the PBLHs at all the 113 radiosonde sites have been successfully derived" and "so have the CALIOP-derived PBLHs" seem the same meaning to me.

(54) line 18-20, there is no verb in this sentence. Also, I do not understand what difference are you talking about? Do you mean "the difference of PBLH derived from CALIOP and from radiosonde"?

(55) line 1-2, I believe you are talking that PBLH exhibit negative values, not sites exhibit negative values. Please correct the description.

(56) line 7-10, I believe the two sentences are expressing the same meanings, please delete one.

(57) line 12-15, please modify it to make it concise.

(58) line 19, occurrence frequency for what?

(59) line 8, 'are' -> 'is'

---

## Author Comment (AC1) · 16 Jun 2016

**Authors' Response to Referees' Comments**

**Anonymous Referee #1:**
Comments on "Planetary boundary layer height from CALIOP compared to radiosonde over China"

**General Comments**

The planetary boundary layer height (PBLH) is an important length scale in weather, climate and air pollution models. The CALIOP-derived PBLHs can construct the PBLH climatology on a global scale. The problem is that the validity of CALIOP-derived PBLH should be examined and the uncertainties of CALIOP-derived PBLH should be known. In this paper, the authors compared the CALIOP-derived PBLH to the radiosonde-derived PBLH in China. The results suggest that they agree very well. The authors also analyzed the difference in the PBLHs derived from the two methods, and showed the spatial distribution of deviations. The results in this paper can help to understand the applicability of CALIOP-derived PBLH in China, and provide the basic information for further investigations. However, some details of the dataset should be further specified, and the English writing should be further improved. Therefore, I recommend the manuscript for publication in ACP, pending minor revisions.

*Response:We are very grateful to referee #1 for his/her positive comments on our work, which are quite constructive and helpful. All of these comments have been explicitly considered and incorporated into this revision. For clarity purpose, here we have listed the reviewers' comments in plain font, followed by our response in italics.*

**Specific Comments**

1. The author declare that the method of Sawyer and Li (2013) was used in this study (in page 6 line 9-10). I suggest that the authors should give a concise introduction of this method, so that the readers can understand how the PBLH is derived from CALIOP in this paper rather than the cited paper. Is this method also applied to the radiosonde data to derive the PBLH? Because the measurement time is almost at noon, the potential temperature profile should exhibit the typical structure of convective BL. Thus the method of maximum potential temperature gradient is suitable for determining the PBLH. Why not use the maximum gradient method? The authors should explain the reason.

*Response:Per your kind suggestions, we gave an concise introduction of this method of Sawyer and Li (2013) in section 2.1 of this revision by adding the following sentences:*

*"By combining wavelet covariance and iterative curve-fitting, Sawyer and Li (2013) developed a novel algorithm (hereafter called SL2013), which can be applied to robustly derive PBLHs from both radiosonde and lidar measurements due to the fact that prior knowledge of instrument properties and atmospheric conditions has been considered. The measurement time of our study is almost at noon, the potential temperature profile more often than not exhibit the typical structure of convective BL. However, due to the potential uncertainties caused by the sensitivity of vertical resolution, and the wide range of sounding time (local time) at different sites across China, SL2013 tends to exhibit advantages over the method of maximum potential temperature gradient. This is most likely because SL2013 is flexible and simple enough for automatic analyses of long-term sounding data at multiple sites, and is able to compensate for noisy signals and low vertical resolution in the soundings. Therefore, SL2013 has been applied to extract PBLHs from radiosonde observations."*

2. The derived PBLH should be the height above the ground. However, shown in Fig. 2, the derived PBLH is above the sea level. Is the terrain height derived from CALIPSO or obtained from other data source? The authors should specify this issue. As shown in Fig. 2, the terrain surface is not very clear in some places.

*Response: We totally agree with you, so we redrew Fig.2 (i.e., Fig.R1 as below). In the figure caption, we described PBLHs as altitude above ground level. The terrain height is directly extracted from CALIOP. Meanwhile, we added in Fig.2 a gray line to better indicate the terrain height clearly.*

[Figure]

*Fig. R1. Curtain plot of attenuated backscatter coefficient as observed from CALIOP aboard CALIPSO on 15 January 2011. The black line indicates the derived PBLH (above ground level) and the grey line immediately on top of the blue region*

*represents the terrain surface (directly from CALIOP data). The red line in the inlet map corresponds to the ground track of CALIOP/CALIPSO over southeastern China.*

3. In page 9 lines 3-4, the authors state "Note that over regions where BL is not convective the retrieved values are not representative of the PBLH (Liu and Liang, 2010)". Also in this section (Section 2.3), the authors describe the method how to eliminate the effects of clouds on the CALIOP-derived PBLH. In other words, the CALIOP data in clear days are used to derive the PBLH, and the BL should be convective. Moreover, the passing time of CALIPSO is 13:30 BJT. Thus it can be expected that the PBLH at this times not very low. However, Table 1 shows that the minimum PBLHs in different seasons are 0.2-0.4 km. I think these values are unbelievable. On the other hand, Table 1 shows that the maximum PBLHs in different seasons are 4-6 km with the largest value in winter. I think these values are also unbelievable. It is likely that uncertainties are introduced in the CALIOP-derived PBLH. Then the problem, to what extent the CALIOP-derived PBLH over China is reasonable, arises. I suggest the author discuss this problem and provide additional information about the statistics of the CALIOP-derived PBLH. For example, by setting the reasonable range of PBLH based on the up-to-date knowledge, the percentage of the derived PBLHs that are in this range can be calculated and compared.

*Response:Thanks for pointing this out. Due to the increasingly polluted atmosphere in China, more stable boundary layers have been frequently observed (e.g., Quan et al., 2013; Gao et al. 2015; Miao et al, 2016). This will inevitably lead to retrieved PBLH values that are not representative of the actual PBLH (Liu and Liang, 2010), even though all the CALIOP data are from 1330 LT overpasses. Also, the large uncertainties are most likely due to the algorithm itself used in extracting CALIOP-derived PBLH. To avoid confusion caused by original Table 1, we added the following description in order to provide more information concerning the statistics of CALIOP-derived PBLH in section 3.2:*
*"As shown in Table 1, we noticed that the maximum PBLHs can reach up to 5-6 km, especially in winter. Therefore, we set the CALIOP-retrieved PBLHs to be within 0.25 and 3km, which seems as a reasonable height range for the midday PBL, highly consistent with the processing methods by McGrath-Spangler (2012). Statistics showed that only 2.1% of all data higher than 3km and 8.8% lower than 0.25km, which have been excluded for further analyses".*

*Reference:*
*Gao, Y, Zhang, M, Liu, Z, Wang, L, Wang, P, Xia, X, Tao, M, Zhu, L.: Modeling the feedback between aerosol and meteorological variables in the atmospheric boundary layer during a severe fog–haze event over the North China Plain. Atmos. Chem. Phys., 15(8): 4279–4295, doi: 10.5194/acp-15-4279-2015, 2015.*
*Liu, S., Liang, X.-Z.: Observed diurnal cycle climatology of planetary boundary layer height. J. Clim., 23, 21, 5790-5809, doi:10.1175/2010jcli3552.1, 2010.*

Miao, Y., Liu, S., Zheng, Y., Wang, S.: Modeling the feedback between aerosol and boundary layer processes: a case study in Beijing, China. Environ. Sci. Pollut. Res., 23(4): 3342–3357, doi: 10.1007/s11356-015-5562-8, 2016.

McGrath-Spangler, E.L., Denning, A.S.: Estimates of North American summertime planetary boundary layer depths derived from space-borne lidar. J. Geophys. Res.--Atmos., 117, 2012.

Quan, J., Gao, Y., Zhang, Q., et al.: Evolution of planetary boundary layer under different weather conditions, and its impact on aerosol concentrations. Particuology. 11(1): 34–40, doi: 10.1016/j.partic.2012.04.005, 2013.

4. For the title of Table 1, "seasonal mean" is not accurate. I think, the maximum PBLH, as well as the minimum PBLH, is not the seasonal mean. Maybe "Statistics of the CALIOP-derived PBLH in different seasons" is more accurate. "Standard deviation PBLH" should be "Standard deviation of PBLH". Moreover, the authors should tell the readers how to determine/calculate the values in the table. Is the maximum/minimum PBLH determined as the maximum/minimum value of one grid in the duration or as the average of the maximum/minimum values at every grid in China? Is the standard deviation calculated at every grid and then averaged in China or calculated directly using all the data?

*Response:Per your suggestions, we clarified the issues pointed out by you and modified the caption of Table 1 as follows:*
*"Table 1. Statistics of the CALIOP-derived PBLH in different seasons during the period 2011 - 2014. The mean PBLHs for all the grids are firstly calculated in China, then the maximum and minimum values of PBLHs are determined by sorting all the mean values. Meanwhile, the mean and standard deviation values of PBLH are determined as the average of mean values at every grid in China."*

5. Following above question, Fig. 8 shows that the CALIOP-derived PBLH ranges from 1.2 km to 2.4km. But the statistics in Table 1 show that the CALIOP-derived PBLH varies in a very large range. How many data are not considered in Fig. 8? The authors should specify this issue in the text or in the figure caption.

*Response: Thanks for pointing this out. We attempt to clarify as follows:*
*In Table 1, all PBLHs derived from CALIOP at every grid across China during the period from 2011 to 2014, which exhibit large variation ranging from 0.15km to 6.13km. However, all the cases with PBLHs greater than 3km or less than 0.25km are viewed as unreliable, which are then removed for further analyses in Fig.8. We have to make sure that PBLHs be extracted simultaneously from both radiosonde and CALIOP observations, leading to less valid collocated data pairs. Moreover, the calculated averaged CALIOP-derived PBLH tends to become more concentrated due to the collocation scheme of the radiosonde measurements and CALIOP, as evidenced in Fig.8. As a consequence, in the caption of Fig. 8, we added the following sentence:*
*"Note that the statistic results are only limited to the samples with collocated CALIOP- and radiosonde-derived PBLHs."*

6. The authors declare in the abstract "The CALIOP observations belonging to Scenario 2 were found to be better for comparison with radiosonde-derived PBLH, owing to smaller difference between them". Similar statements are found in the conclusion section. However, Fig. 7 shows that the mean difference for Scenario 3 is the smallest. What is the solid evidence for this conclusion?

*Response:In order to find more solid evidence to support the argument, we added to the revised manuscript one new figure (Figure 8, i.e., Figure R2 here), which shows the calculated 5th, 25th, 50th, 75th and 95th percentile values of PBLHs derived from CALIOP and radiosonde for each scenario. As such, to get a comprehensive understanding of the differences existing among various scenarios, the following texts have been added to section 3.4:*

*"As indicated in Figure 8, Scenario 2 witnesses the least difference of 0.08km between the CALIOP- and radiosonde-median PBLH values in contrast to larger differences of 0.24km and 0.12km for Scenario 1 and Scenario 3, respectively. In addition, the PBLH differences in terms of 25th and 75th percentile values for Scenario 2 are much more indiscernible, as compared with those for other two scenarios. This implies that Scenario 2 gains more advantages over other two scenarios due to the smaller difference between CALIOP- and radiosonde-derived PBLHs."*

[Figure]

*Fig. R2. Box-and-whisker plot showing the 5th, 25th, 50th, 75th and 95th percentile values of PBLH derived from CALIOP (in blue) and radiosonde (in red) for each scenario. Note that only 1400 BJT radiosonde are used to make comparison with afternoon CALIOP-derived PBLHs.*

**Technical Corrections**

(1) The grammatical errors should be corrected (Just some are listed here. The author should thoroughly check for simple typos and grammatical errors). For example,

Page 2 line 1, "for comparison with" should be "in comparison with".

Page 2 line 2, "at early summer afternoon" should be "in early summer afternoon".

Page 3 line 20, "the fact the number" should be "the fact that the number".

Page 4 line 22, "are" should be "is".

Page 6 line 9, "this methods" should be "this method".

Page 8 line 7, "in combination with and" should be "in combination with".

*Response: Except for the typos as you pointed out here, other grammatical errors have been corrected in this revision.*

(2) Fig. 2, at the top of this figure the times "05:33:17" and "05:47:14" should be the local times "13:33:17" and "13:47:14".

*Response:Per your kind suggestions, the time at the top of Fig.2 has been changed to the local times, i.e., "13:33:17 (BJT)" and "13:47:14 (BJT)".*

(3) Fig. 7, the value of mean difference between the CALIOP-and radiosonde-derived PBLHs in each panel (0.17km, 0.22km, 0.17km and 0.15km respectively). But the figure shows that the difference for a single site is either positive or negative (denoted by different colours). How to calculate the mean value, directly or by the absolute values? I guess by absolute values. Therefore the absolute value sign should be added to ΔPBLH.

*Response:We appreciate you pointing it out. You are right, the difference of PBLH was supposed to denote absolute value. Therefore, it has been changed to "|ΔPBLH|" in Fig. 7.*

---

## Author Comment (AC2) · 16 Jun 2016

**Authors' Response to Referees' Comments**

**Anonymous Reviewer #2:**

Comments on "Planetary boundary layer height from CALIOP compared to radiosonde over China"

The planetary boundary layer height (PBLH) is an important parameter for the weather and climate study, as well as atmospheric pollution study. This study tries to obtain global PBLH based on CALIPSO satellite observations, and carried out an intercomparison study with those from radiosondes and lidars here. The results suggest that they agree reasonably well in China regions. This is a valuable contribution to the science community to better understand the potential applicability of CALIPSO observations to obtain PBLH. However, this paper does need some improvement as detailed below, particularly regarding to the English writing. I would recommend the manuscript for publication in ACP, pending minor revisions.

*Response:We are quite grateful to referee #2 for his/her positive comments on our work, which are quite constructive and helpful. All these comments and concerns raised by the referee have been explicitly considered and incorporated into this revision. For clarity purpose, here we have listed the reviewers' comments in plain font, followed by our response in italics.*

**Main Comments**
1. The English writing strongly need improve. The paper descriptions could be more concise and accurate.
*Response:Per your kind suggestions, we have improved the English writing, both grammatically and scientifically. Meanwhile, the descriptions have been revised to be as concise and accurate as possible in this revised manuscript.*
2. One key role of this study as the author expressed is "The PBLH retrieval from CALIOP is expected to complement the ground-based site measurement due to its large spatial coverage". However, I think the pass of CALIPSO satellite over a specific location is limited. May you please provide more information about the CALIPSO passed regions?
*Response:We agree with the reviewer that the pass of CALIPSO satellite over a specific location is temporally limited (especially in the capability of charactering diurnal variation of PBL). As shown in Figure 1, during one CALIPSO revisit cycle (16 days), there are about 42 ground tracks in China for the daytime ascending overpasses (1330 LT). And the neighboring ground tracks of CALIPSO are in the intervals of approximately 100-150 km, depending on latitudes. To make the*

*description more accurate, in the introduction section, we added "From the climatological point of view" just before "the PBLH retrieval from CALIOP is expected to complement the ground-based site measurement due to its large spatial coverage."*

3. Section 2.1, I would like to know the uncertainties in the PBLHs obtained from radiosondes, which is very important since the authors are using them to evaluate those from CALIOP.

*Response: The uncertainties associated with PBLH obtained from radiosonde come from (1) the estimation methods of PBLH, which are generally referred to structural uncertainty (Seidel et al., 2010). To our knowledge, the method (Sawyer and Li, 2013) we used here is one of the most advanced algorithms, in which prior knowledge of instrument properties and atmospheric conditions has been adequately taken into account; (2) the extreme adverse weather, which is also an important influential factor. For instance, the PBL as deep convective cloud occurs will collapse, leading to an extremely large value; (3) the failed launch of weather balloon. All of these uncertainties have been reflected in this revision.*
*Reference:*

*Seidel, D.J., Ao, C.O., Li, K.: Estimating climatological planetary boundary layer heights from radiosonde observations: Comparison of methods and uncertainty analysis. J. Geophys. Res. -Atmos. 115, 2010.*

4. Section 2.2, what is the uncertainties of PBLHs from lidars, and what are the extra uncertainties caused by the selection of compare region size?
*Response:In our points of view, the uncertainties of PBLHs from lidars largely come from the contamination caused by boundary layer cloud, along with the heavy haze which always leads to strong signal attenuation.*

*Moreover, the temporal window utilized to take averages centered at the observation time of ground-based lidar may be a factor influencing the PBLH uncertainty. To just name a few, the thorough analysis by Hennemuth and Lammert (2006) indicated that 10-min window leads to an average bias of 150 m as compared with 1-h window. All of these uncertainties have been discussed in detail and reflected in the last paragraph in section 2.2 of this revised manuscript.*

*To make the intercomparison more robust, a circle with a radius of 75 km centered at ground site was chosen to obtain averaged PBLH from CALIOP. As such, at least 100 samples around each radiosonde site can be used for the estimation of PBLH from CALIOP, given the 5km resolution along CALIPSO track.*
*Reference:*
*Hennemuth B, Lammert A. Determination of the atmospheric boundary layer height from radiosonde and lidar backscatter [J]. Boundary-Layer Meteorology, 2006, 120(1): 181-200.*

5. Section 3.1, this is a comparison. If you would like to say 'evaluation", you need assume the accuracy of ground-based lidar-derived PBLH with at least clear uncertainty information.
*Response:Per your kind suggestion, "evaluation" has been changed to "comparison".*

6. Section 3.2, I would suggest you add the climatology of PBLH from the radiosonde profiles over China and compare this with your results from CALIPSO observations. This could let us know how reliable of your CALIPSO-derived PBLHs.

*Response:Per your suggestion, the climatology of PBLH from the radiosonde profiles over China was added, as shown in Fig. R3 (i.e., Figure S2 in the supplementary material). Note that only the radiosonde-derived PBLH climatology at 1400 BJT in summertime is and should be used for comparison with CALIOP-derived PBLHs. In order to let the readers better know the reliability of CALIOP-derived PBLHs, the following description was added in the first paragraph of section 3.4:*

*"In terms of the spatial differences of PBLHs, both CALIOP retrievals (Figure 4b) and radiosonde observations (Figure S2) show that large PBLH values tend to occur at Tibetan Plateau, southwestern China, and northern China in early summer afternoon. This is likely indicative of good agreement between CALIOP- and radiosonde-derived PBLH retrievals"*

[Figure]

*Fig. R3. Spatial distribution of climatological PBLHs derived from radiosonde at 1400 BJT in summer (June-July-August, JJA) during the period from 2011 to 2014.*

**Specific Comments:**
Page1
(1) Line 12: The description could be more concise: the accurate estimation of planetary boundary layer height (PBLH) …. The PBLH retrieved from …"
*Response:Amended as suggested.*
(2) Line 17: ground-based and satellite-based or ground-based and spaceborne.
*Response:Amended as suggested.*
(3) Line 17-18, for r=0.59 or 0.65, could we say "good agreement"?
*Response:The sentence has been revised to "Comparison between PBLHs from ground- and satellite-based lidars leads to a correlation coefficient of 0.59 in Beijing and 0.65 in Jinhua, respectively."*

(4) Line 19, 'during 2011 to 2014' -> 'for the period from 2011 to 2014'

*Response:Amended as suggested.*

(5) Line 19, lower values

*Response:Amended as suggested.*

(6) What is the uncertainty for PBLH from radiosonde observations? What are the factors that could result in the differences in PBLH between satellite-and ground-based observations, and their contributions?

*Response:Please see our response to main comment #3.*

Page2

(7) Line 17, how do you arrange the order of references?

*Response:We rearranged the order of references to chronological order by year of publication, which shows as follows: "(Medeiros et al., 2005; Hong et al., 2006; Zhang et al., 2007; Hu et al., 2010)."*

(8) Line 18-20, the sentence have grammar error with 2 verbs.

*Response: The sentence you pointed out has been revised as follows:*

*"The depth (or height) of PBL, which determines the vertical extent of turbulent mixing and convection activity within it, is a key length..."*

(9) line 1-3, why is it required 4-8 times for IOP experiment?

*Response:Generally speaking, 4-8 times are required during IOP experiment to better capture the diurnal variation in the thermodynamic and dynamic conditions of atmosphere.*

(10) line 4, how accurate of the PBL height is it for the measurements from radiosondes?

*Response:Please see our response to question 3 for more detail.*

(12) line 12-13, what do you mean with (Amiridis et al.) in these lines? Reference?

*Response:It means reference. Therefore, we added a reference"(Seibert, 2000)" here.*

(13) line 13-15, what do you mean for this sentence: "large seasonal and diurnal variations in PBLHs were observed between the different methods applied to radiosonde, ground-based lidar, CALIOP observations over one site in South Africa"

*Response: It has been changed to "large seasonal and diurnal variations in PBLHs were observed, most likely due to the different methods utilized to…"*

(14) what do you mean for "large scale land-based observations"?

*Response: We clarified it by changing it to "large scale ground-based radiosonde observations" in this revision.*

(15) how reliable for the ground-based lidar observation of PBLH?

*Response:Please see the response to main comment # 5 for more details.*

(16) line 14, times -> time

*Response:Amended as suggested.*

(17) line 15, why call the summer as flood season? It might be wet season, but not good as flood season?

*Response: "flood season" has been changed to "wet season".*

(18) line 16, what do you mean for "severe weather forecasting"?

*Response:The sentence has been changed to "CMA required the soundings to be launched three to four times a day in summer (the wet season), i.e., 0200 BJT, 0800 BJT, 1400 BJT, and 2000 BJT to seamless monitor the vertical structure of atmosphere, and thus to better serve the high-impact weather forecasting."*

(19) line 16-19, 'owe to ..., ... therefore..."?

*Response: "therefore" was removed .*

(20) line 9, What are you comparing to regarding "a good agreement"?

*Response:We rewrote the sentence as follows:*

*"By combining the methods of wavelet covariance and iterative curve-fitting (Steyn et al., 2009), Sawyer and Li (2013) developed a novel algorithm (hereafter called SL2013), which can be applied to robustly derive PBLHs from both radiosonde and lidar measurements due to the fact that prior knowledge of instrument properties and atmospheric conditions has been adequately considered."*

(21)line 9, 'this methods of ... was ...'?

*Response:"methods" has been changed to "method".*

(22) line 6, 'the algorithm in Zhang et al. (2015) are applied on ..."-> "the algorithm developed by Zhang et al. (2015) are applied to ..."*

*Response:Amended as suggested.*

(23) line 7, what kind of profiles are you talking about? lidar profiles?

*Response:We are referring to CALIOP profiles.*

(24) line 8-9, why do you choose the area with radius of 75 km?

*Response:See our response to main comment #3, please.*

(25) line 10-13, what are the data volume fraction for these cases?

*Response:Overall, the data volume fraction is roughly 87.7 %. To better describe the ground-based lidar data, we added Figure R4 (i.e., Figure S1 in the supplementary material). The related description was added to the end of section 2.2.*

[Figure]

*Fig. R4. Statistics showing the fractional volumes (in percent) of lidar measurement at Beijing during the whole year of 2014 stratified by no observation (in red), without PBLH retrievals due to weather conditions (in yellow), and with PBLH retrievals (in green).*

(26) line 17-19, please correct the sentences, such as "It measures attenuated backscatter coefficients at resolutions of 1/3 km in the horizontal and
30 m in the vertical at the visible wavelength ..."

*Response:The sentences have been changed to "It measures attenuated backscatter coefficients at a resolution of 1/3 km in the horizontal at the visible wavelength (532 nm) and near-infrared wavelength (1064 nm), and its vertical resolution varies with altitude (h): 30m from ground up to h = 8.2 km, 60m from h = 8.2 km to 20.2 km, and 180m from h = 20.2 km to 30.1 km (Winker et al.,2009; Huang et al.,2015)"*

(27) line 7, " in combination with and ..."?

*Response:It has been changed to "in combination with.." .*

(28) line 8-9, "This is because that ...", You do not need to explain since you have said for "cloud screening"

*Response:The redundant sentence you pointed out has been removed according to your kind suggestion.*

(29) line 9-11, please indicate the advantage of your choosing method.

*Response:Just following "..be inferred (McGrath-Spangler and Denning, 2012, 2013)." The following sentence was added: "However, either maximum variance algorithm or Haar wavelet technique has its weakness due to the strong dependence on the chosen strategy in the threshold values."*

(30) line 11, there are two periods.

*Response:One redundant period was removed.*

(30) line 9-16, please tell readers the uncertainties or the uncertainty-influential factors for this determination method.

*Response:We added the sentence as follows: "However, either maximum variance algorithm or Haar wavelet technique has its weakness due to the strong dependence on the chosen strategy in the threshold values."*

(31) line 16-19, this is redundant since you have mentioned the 75 km earlier. Also, why do you select 75 km, not 50 or 25 km?

*Response:These redundant sentences have been removed, and the following paragraph was added to the end of 2nd paragraph in section 2.2:*

*"Due to the neighboring ground tracks of CALIPSO at approximately 100-150 km longitudinal interval over China, a 75km-radius circle centered at each ground-based lidar site has been determined for its spatial matchup with CALIOP, so has the matchup of radiosonde site with CALIOP."*

(32) line 1, what do you mean "valid" here? For the overpasses, are there invalid ones? I do not understand.

*Response*:*"valid" means without cloud. Therefore, we modified the sentence to "The CALIPSO measurements were retained for PBLH retrievals at grid points where the number of valid (i.e., without cloud)…"*

(33) line 4, How do you determine if the BL is convective or not?

*Response*:*Our method utilized in PBLH retrieval (see our response to general comment #1 by reviewer #1 for details) does not rely on whether the BL is convective or not, and thus the sentence was deleted in this revision.*

(34) line 5-10, you just gave one case to show the good agreement between two algorithms (even17 profiles averaged within a 5 km region). This is not enough to conclude that "the combined algorithms are reliable".

*Response: The sentence of "indicating that the combined algorithms is reliable " was deleted in this revision.*

(35) line 10, 'is' ->'are'

*Response*:*Amended as suggested.*

(36) line 13,are you sure your comparison study is "a first attempt"?

*Response*:*We deleted "a first attempt" and revised the sentence to "In order to make the intercomparison more reliable between CALIOP- and radiosonde-derived PBLHs…".*

(37) line 15-16, how do you exclude the cases with cloud cover? In other words, how do you get the cloud coverage?

*Response*:*The cases were manually determined whether they were contaminated or not, based on the meteorological data from the neighboring weather station.*

(38)line 17, "shows that"? I believe it should be just "shows"

*Response*:*You are right, and thus "that" was deleted as suggested.*

(39) line 17-21, for so limited data samples, how reliable are the comparison results?

*Response*:*We rewrote these sentences as below:*

*"Due to the samples being still limited, we cannot be quite sure to argue that the CALIOP-derived PBLHs are reliable enough. Further evaluation studies are warranted in the future as long as more ground-based lidar observations are available. However, the correlation coefficients obtained here are similar to those reported at SACOL site of northwestern China (e.g., Liu et al., 2015)."*

(40) line 1-2, the correlation coefficients are low, why do you say 'show a good agreement'?

*Response*:*"which shows a good agreement" was deleted.*

(41) line 11-13, the variability in winter (0.4 km) is larger than that in summer (0.31 km), why do you say the lowest PBLH variability occurs in winter?

*Response*:*Per your suggestion, the "variability" has been removed, and the sentence has been changed to "the lowest PBLH values occur in winter".*

(42) line 13, "were occurred" -> "occur'

*Response*:*Amended as suggested.*

(43) line 14-15, please modify the description to make it more concise.

*Response*:*We modified the sentence as follows:*

*"…when the development of PBL is typically suppressed due to the less solar radiation received at the surface. In contrast, the more intense solar radiation reaching the surface in summer favors the development of PBL (Stull et al., 1988)."*

(44) line 19, 'was' -> 'were'

*Response:Amended as suggested.*

(45) line 21, 'may be suppressed by aerosol radiative effects and aerosol-wind interactions(Xia et al., 2007; Yang et al., 2016)'

Yang, X., C. Zhao, J. Guo, Y. Wang, 2016, JGR: intensification of air pollution associated with its feedback with surface solar radiation and winds in Beijing,

*Response:Amended as suggested.*

(46) line 2, 'had been' -> "have been"

*Response:Amended as suggested.*

(47)line 5-7, this information has been described two times earlier. I would suggest a more detailed description for only one time.

*Response:We can not agree with the reviewer any more, so we deleted it in the first paragraph of section 3.3, and more detailed description concerning the matchup scheme between radiosonde and CALIOP was added in section 2.3.*

(48)line 7-9, this also seems redundant.

*Response:It has been deleted as suggested.*

(49) line 14, delete "On the other hand,"

*Response:Deleted.*

(50)line 16, 'can be' -> 'are'

*Response:Amended as suggested.*

(51) line 8, what do you mean for "basically"?

*Response:"basically" has been revised to "mostly".*

(52) line 11-12, could you give me a little more explanation? I do not understand the logic here.

*Response:We have revised the sentences as follows:*

*"..The more northward the radiosonde sites, the greater number of the CALIPSO overpasses over the same circle of 75 km radius. Therefore, the distinct discrepancy in geographic distributions of radiosonde sites belonging to Scenarios 1 and 3 are most likely due to the latitude differences…"*

(53) line 16-17, "the PBLHs at all the 113 radiosonde sites have been successfully derived" and "so have the CALIOP-derived PBLHs" seem the same meaning to me.

*Response:We have revised the sentence to "Using the algorithms as detailed in Section 2, the PBLHs at all the 113 radiosonde sites have been successfully derived from radiosonde and CALIOP."*

(54) line 18-20, there is no verb in this sentence. Also, I do not understand what difference are you talking about? Do you mean "the difference of PBLH derived from CALIOP and from radiosonde"?

*Response*:*You are right, and thus we revised the sentence to: "..the differences of PBLHs at every radiosonde sites (Figure 1) from CALIOP measurements at 1330 LT minus those from radiosonde observations at 1400 BJT in the summertime (June-July-August) during the period of 2011-2014 are calculated..."*

(55) line 1-2, I believe you are talking that PBLH exhibit negative values, not sites exhibit negative values. Please correct the description.

*Response*:*Per your kind suggestion, we changed the sentence to "As shown in Figure 7(a), the PBLH differences over most of the radiosonde sites .."*

(56) line 7-10, I believe the two sentences are expressing the same meanings, please delete one.

*Response*:*Per your kind suggestion, we deleted "Note that we cannot totally rule out other factors that may also contribute to the east-west gradient."*

(57) line 12-15, please modify it to make it concise.

*Response: It has been shortened as "…Overall, the radiosonde-derived PBLHs tend to be overestimate compared with CALIOP-derived PBLHs due to the majority of radiosonde sites…"*

(58) line 19, occurrence frequency for what?

*Response*:*Occurrence frequency for the number of radiosonde sites*

Page 15    (59) line 8, 'are' -> 'is'

*Response*:*Amended as suggested.*

---

## Author Comment (AC3) · 17 Jun 2016

The comment was uploaded in the form of a supplement:
http://www.atmos-chem-phys-discuss.net/acp-2016-250/acp-2016-250-AC3-supplement.pdf

---

## Referee Report (RR1)

**Comments on "Planetary boundary layer height from CALIOP compared to radiosonde over China"**

**General Comments**

The planetary boundary layer height (PBLH) is an important length scale in weather, climate and air pollution models. The CALIOP-derived PBLHs can construct the PBLH climatology on a global scale. In this paper, the authors compared the CALIOP-derived PBLH to the radiosonde-derived PBLH in China. The results suggest that they agree very well. The authors also analyzed the difference in the PBLHs derived from the two methods, and showed the spatial distribution of deviations. These results can help to understand the applicability of CALIOP-derived PBLH in China, and provide the valuable information for further investigations. This version of manuscript is substantially improved and the results are presented more clearly than in the original one. I recommend the manuscript for publication in ACP, pending minor revisions.

**Specific Comments**

The revisions are not specific, but represent a general need to improve the English wording and writing. Many sentences in this version do not read smoothly. I suggest the authors thoroughly check their document. The authors had better get a fluent writer/speaker of English to look through the paper before submitting the final version.

---

## Author Response (AR2)

**Authors' Response to Co-editor and Referees' Comments**

**Co-Editor Decision:**

5  Publish subject to technical corrections (08 Jul 2016) by Prof. Aijun Ding

**Comments to the Author:**

Both reviewers are satisfied with revision and would like to recommend the publication of the paper in Atmos. Chem. Phys., however one review suggested a polish of English writing before
10  publish. I would like to suggest the author to ask a native speaker to check the paper throughout. In addition to this, I would like to suggest the author add few sentences to link the paper to PEEX in introduction or discussion part, because this paper will be published in PEEX special issue other than a normal ACP paper.

*Response: We are quite grateful to Prof. Aijun Zhang and both referees for their positive*
15  *comments on our work, which are quite constructive and helpful. All these comments and concerns raised by the referee have been explicitly considered and incorporated into this revision. For clarity purpose, here we have listed the co-editor and referees' comments in plain font, followed by our response in italics. Per your kind suggestion, we have asked native English speaker to help polish the editing of this paper throughout. In addition, we added one*
20  *paper by Kulmala et al. (2015) to link this manuscript to PEEX in the section of Introduction.*

**Referee #2:**

The planetary boundary layer height (PBLH) is an important length scale in weather, climate and air pollution models. The CALIOP-derived PBLHs can construct the PBLH climatology on a global scale. In this paper, the authors compared the CALIOP-derived PBLH to the radiosonde-derived PBLH in China. The results suggest that they agree very well. The authors also analyzed the difference in the PBLHs derived from the two methods, and showed the spatial distribution of deviations. These results can help to understand the applicability of CALIOP-derived PBLH in China, and provide the valuable information for further investigations. This version of manuscript is substantially improved and the results are presented more clearly than in the original one. I recommend the manuscript for publication in ACP, pending minor revisions.

**Specific Comments**

The revisions are not specific, but represent a general need to improve the English wording and writing. Many sentences in this version do not read smoothly. I suggest the authors thoroughly check their document. The authors had better get a fluent writer/speaker of English to look through the paper before submitting the final version.

*Response:We appreciate the positive comments on our work, which are quite constructive and helpful. Per your suggestion, we have asked native English speaker to help polish the editing of this paper throughout, please see the attached marked-up manuscript for details.*

**Planetary boundary layer height from CALIOP compared to radiosonde over China**

Wanchun Zhang[1], Jianping Guo[1], Yucong Miao[1,2], Huan Liu[1], Yong Zhang[3], Zhengqiang Li[4], Panmao Zhai[1]

[1]State Key Laboratory of Severe Weather, Chinese Academy of Meteorological Sciences, Beijing 100081, China
[2]Department of Atmospheric and Oceanic Sciences, Peking University, Beijing 100871, China
[3]Meteorological Observation Centre, China Meteorological Administration, Beijing, 100081, China
[4]State Environmental Protection Key Laboratory of Satellites Remote Sensing, Institute of Remote Sensing and Digital Earth of Chinese Academy of Sciences, Beijing 100101, China

*Correspondence to*: Drs Jianping Guo (jpguocams@gmail.com) and Panmao Zhai (pmzhai@cma.gov.cn)

**Abstract.** Accurate estimation of planetary boundary layer height (PBLH) is key to air quality prediction, weather forecast, and assessment of regional climate change. The PBLH retrieval from CALIOP is expected to complement ground-based measurements due to the broad spatial coverage of satellites. In this study, CALIOP PBLHs are derived from combination of Haar wavelet and maximum variance techniques, and are further validated against PBLHs estimated from ground-based lidar at Beijing and Jinhua. Correlation coefficients between PBLHs from ground- and satellite-based lidars are 0.59 at Beijing and 0.65 at Jinhua, respectively. Also, the PBLH climatology from CALIOP and radiosonde are compiled over China during the period from 2011 to 2014. Maximum CALIOP-derived PBLH can be seen in summer as compared to lower values in other seasons. Three matchup scenarios are proposed according to the position of each radiosonde site relative to its closest CALIPSO ground tracks. For each scenario, inter-comparisons were performed between CALIOP- and radiosonde-derived

已删除: The a
已删除: so on
已删除: the
已删除: site
已删除: its
已删除: large
已删除: To such end,
已删除: are
已删除: estimated from CALIOP, using uses the
已删除: which
已删除: then
已删除: Comparison between
已删除: leads to
已删除: shows a correlation coefficient of
已删除:
已删除: in
已删除: in

[revised manuscript text omitted]